# Cohort profile: the U-BIRTH study on peripartum depression and child development in Sweden

Hsing-Fen Tu [1,2] Emma Fransson,[1,3] Theodora Kunovac Kallak,[1] Ulf Elofsson,[1] Mia Ramklint,[4] Alkistis Skalkidou[1]

[1]Department of Women's and Children's Health, Uppsala University, Uppsala, Sweden
[2]Department of Psychology, Uppsala University, Uppsala, Sweden
[3]Centre for Translational Microbiome Research, Department of Microbiology, Tumor and Cell Biology, Karolinska Institutet, Stockholm, Sweden
[4]Department of Medical Sciences, Uppsala University, Uppsala, Sweden

**Correspondence to**
Dr Alkistis Skalkidou;
Alkistis.Skalkidou@kbh.uu.se

## ABSTRACT

**Purpose** The current U-BIRTH cohort (Uppsala Birth Cohort) extends our previous cohort Biology, Affect, Stress, Imaging and Cognition (BASIC), assessing the development of children up to 11 years after birth. The U-BIRTH study aims to (1) assess the impact of exposure to peripartum mental illness on the children's development taking into account biological and environmental factors during intrauterine life and childhood; (2) identify early predictors of child neurodevelopmental and psychological problems using biophysiological, psychosocial and environmental variables available during pregnancy and early post partum.

**Participants** All mothers participating in the previous BASIC cohort are invited, and mother–child dyads recruited in the U-BIRTH study are consecutively invited to questionnaire assessments and biological sampling when the child is 18 months, 6 years and 11 years old. Data collection at 18 months (n=2882) has been completed. Consent for participation has been obtained from 1946 families of children having reached age 6 and from 698 families of children having reached age 11 years.

**Findings to date** Based on the complete data from pregnancy to 18 months post partum, peripartum mental health was significantly associated with the development of attentional control and gaze-following behaviours, which are critical to cognitive and social learning later in life. Moreover, infants of depressed mothers had an elevated risk of difficult temperament and behavioural problems compared with infants of non-depressed mothers. Analyses of biological samples showed that peripartum depression and anxiety were related to DNA methylation differences in infants. However, there were no methylation differences in relation to infants' behavioural problems at 18 months of age.

**Future plans** Given that the data collection at 18 months is complete, analyses are now being undertaken. Currently, assessments for children reaching 6 and 11 years are ongoing.

## INTRODUCTION

Peripartum depression, a significant complication that affects many women during pregnancy and/or the first year post partum,[1] affects one in 4–7 pregnancies[2–5] and poses a serious global public health concern. Peripartum depression heightens the risk of

## STRENGTHS AND LIMITATIONS OF THIS STUDY

⇒ This population-based study provides rich data across pregnancy, infancy, early childhood, and the pre-teen periods; the sample corresponds to one of the largest cohorts within the Scandinavian countries.
⇒ To disentangle how the complex interactions between maternal well-being and health, peripartum exposures and childhood environment contribute to the cognitive, social and emotional development of the child, this study includes data from biological, psychosocial and behavioural assessments using a longitudinal design.
⇒ The cohort's representability is relatively low at about 10% of the background population, impacting on the generalisability of the findings.
⇒ Though the assessments of child development included in the project are well accepted and widely applied in the field, most of them rely mainly on parent-report questionnaires, which could introduce information bias.

maternal self-harm and suicidal acts[6 7] and has a substantial and long-term impact during the most vulnerable window of opportunity for the human brain development.[8] Exposure to peripartum depression during this critical period might result in lasting physical, social–emotional, neurological, and cognitive problems in children,[9–13] even into adulthood.[14] However, while the long-term associations between peripartum mental health and child outcomes are well documented,[15] the underlying mechanisms remain less understood. Given the high degree of complexity and the non-linear association between the trajectories of maternal depression and children's development,[16 17] it is challenging to study factors deeply embedded in the temporal, social, cultural, biological, genetic and physical environment. The U-BIRTH cohort (Uppsala Birth Cohort), builds on the Biology, Affect, Stress, Imaging and Cognition (BASIC) study,[18] aiming to provide an in-depth understanding of the relationships

between types of peripartum depression and the developmental trajectories of children, as well as the possible mechanisms of cross-generational transmission. The BASIC study explored various maternal factors linked to peripartum depression, enhancing our understanding of the mechanisms. The combination with BASIC study and the U-BIRTH cohort study will aid in identifying individual risk profiles and support personalised early prevention and intervention programmes.

Peripartum depression affects both mothers and children daily life and elevates the risk of the children for poor self-regulation,[10] delayed executive functions[11] and neurodevelopmental disorders, such as attention deficit hyperactivity disorder and autism spectrum disorder.[12 13] Meanwhile, mothers with peripartum depression might have a higher risk for chronic maternal depression.[16] Children of depressed mothers are at risk of various problems in face challenges in social, cognitive and academic functioning compared with those of non-depressed mothers.[15 19–21] To capture the full path of the cross-generational transmission of child development, the identification of early risk factors for peripartum depressive symptoms has been an important topic in the field. Previous research has shown potential factors like inflammatory biomarkers (eg, increased levels of interleukin (IL)-6, IL-8, C-reactive protein (CRP) and tumour necrosis factor-α (TNF-α), see review by Silva-Fernandes et al[22]), genetic factors (eg, serotonin transporter gene (5-HTT), catechol-O-methyl-transferase, mono-amine-oxidase type A and so on),[23] and psychosocial stressors (eg, high life stress, lack of social support, history of depression, and so on) for peripartum depression.[24–28] It is essential to further elucidate the longitudinal and dynamic relationship between these potential risk factors, maternal depressive symptoms and child development.

The U-BIRTH study, drawing on life course epidemiology concepts and the significance of early life factors in long-term health,[29] sets out to study the impact of maternal depressive trajectories on children's social and emotional development. In summary, the U-BIRTH study thus aims to (1) assess the impact of exposure to peripartum mental illness on children's development taking into account biological, psychological and environmental factors during intrauterine life and childhood; (2) identify early predictors of child neurodevelopmental and psychological problems using biophysiological, psychosocial and environmental variables available during pregnancy and early post partum. Specific objectives of the current protocol include: (1) presenting a detailed description of the U-BIRTH study, a continuation of the BASIC cohort with follow-ups up to 11 years after delivery, and (2) giving updated summaries of the previous findings of associations between peripartum mental health and offspring's outcomes from both the BASIC and U-BIRTH studies. Ultimately, we hope to identify predictors as well as risk and protective factors that might support future prevention and intervention programmes.

## COHORT DESCRIPTION
### Study design
The U-BIRTH study, an extension of the BASIC study,[18] is hosted at the Department of Obstetrics and Gynaecology at Uppsala University Hospital. The BASIC study (from September 2009 to November 2018) involved 5492 women (in 6478 pregnancies, ca. 22% of all Uppsala County births). During the BASIC study recruitment, all Swedish-speaking women aged 18 years and older with a routine ultrasound at Uppsala University Hospital were invited to participate. Exclusion criteria included age under 18 years, inadequate Swedish proficiency, protected identities, bloodborne infections and/or non-viable pregnancy detected through routine ultrasound. These criteria remain the same in the U-BIRTH study. Women received written information by post and provided their written consent for modalities they wished to take part in (eg, questionnaires, blood samples, tissue samples, or microbiota samples). Participating women were followed up from gestational weeks 16–18 through 12 months post partum. In the BASIC study,[18] data were collected using various psychometric self-report instruments and biological samples (more information: https://www.basicstudie.se/). For the U-BIRTH study (https://www.ubirth.se/), initiated in 2012, all the BASIC study participants not opting out in any way were (will be) invited when their children reached (or reach) 18 months, 6 and 11 years of age. The youngest participants were born in April 2019. Data acquisition for 6 years old and 11 years old is expected to finish in the summer 2025 and 2030. Prior to each time point for assessments, consent from all participating mother–child dyads is (will be) obtained. Many participating mother–child dyads already have data from multiple time points acquired within the BASIC study, with the sociodemographic information available in table 1. In the U-BIRTH study, all recruited mother–child dyads are invited to a series of assessments when the child is at age of 18 months, 6 years and 11 years (table 2). Figure 1 illustrates the data timeline included in the protocol. For a full list of assessment tools for mothers and children in both BASIC and U-BIRTH studies, see online supplemental table 1. Currently, 3525 families have completed at least 1 assessment and 52 mother–child dyads dropped out due to external factors (eg, death of the mother or the child or the family moved away). Overall, all children have reached 18 months of age and the data acquisition of this phase is completed. There were 6083 invitations to the study for the 18 months post partum, and 3373 mother–infant dyads (55%) consented to participate. Among those consenting to participate in the survey, 85% completed the survey at 18 months post partum. Seventy-three per cent of the children have today reached 6 years of age. There were 4340 invitations sent, and 1409 (32%) mother–child dyads have fully completed the assessments. Moreover, 1547 (25%) children have reached 11 years of age and 624 have completed the assessments. Families not responding to the invitation receive one email reminder. Families who agree to participate but do

**Table 1** Sociodemographic information of the U-BIRTH cohort and the general population of Uppsala County and Sweden

| Variable | U-BIRTH (n=3525*) | | | Uppsala county† (2009–2016, n=32 258) | Sweden† (2009–2016, n=902 698) |
|---|---|---|---|---|---|
| | Available data (n) | % or mean | Missing, n | % or mean | % or mean |
| Maternal age (mean, SD) | 3374 | 31.8 (±4.4) | 151 | 30.4 (±5.2) | 30.3 (±5.3) |
| Country of origin | | | | | |
| Scandinavia | 3069 | 87.1 | 227 | 80.3 | 76.6 |
| Other countries | 229 | 6.5 | | 19.7 | 23.3 |
| Education | | | | | |
| ≤ 12 years | 520 | 17 | 471 | 46.7 | 48.5 |
| Cohabiting with the second legal guardian | 2790 | 97.2 | 656 | 95.7 | 93.1 |
| BMI before pregnancy kg/m$^2$ | 3094 | 23.7 (±4.2) | 431 | | |
| < 18.5 | 93 | 2.6 | | 2.2 | 2.9 |
| 18.5≤BMI < 25 | 2146 | 60.9 | | 58.6 | 65.6 |
| 25≤BMI < 30 | 594 | 16.9 | | 25.9 | 25.3 |
| ≥ 30 | 261 | 7.4 | | 13.4 | 13.0 |
| Infant birth weight, gram | 3257 | 3593 (±546.6) | 268 | 3498 | 3504 |
| Infant birth weight<2500 g | 92 | 2.8 % | | 5.1 | 4.4 |
| Infant birth weight≥4500 g | 142 | 4.4 | | 3.9 | 3.5 |
| Primiparous | 1547 | 47.5 | 265 | 43.8 | 43.9 |
| Premature delivery (< week 37) | 168 | 5.2 | 265 | 7.4 | 6.0 |
| Apgar 5<7 | 40 | 1.2 | 282 | 2.0 | 1.3 |

Percentages are given in relation to available data from women.
*Data downloaded on 2023-01-27. At this time points, there were 3525 families already completed at least one time assessment. These children were all born between January 2010 and April 2019.
†Data retrieved on 27 January 2023 from the Swedish Medical Birth Register and Swedish National board of Health and Welfare (www.socialstyrelsen.se).
BMI, body mass index.

not answer the survey receive three email reminders. The flowchart for participants is presented in figure 2. For those who decided to drop out without any external cause (eg, death of the child or the mother), some provide reasons such as moving away, while others did not specify. For non-respondents, we send a reminder but cannot ascertain the reasons.

### Data collection
In the U-BIRTH, three follow-up timepoints include mother-reported instruments and saliva sampling from children at 18 months and 6 years of age. For detailed measures and timeline, see table 2 and figures 1 and 2 (a full list covers information in both BASIC and U-BIRTH, see online supplemental table 1). Mothers are also invited to complete the Patient Health Questionnaire, at ages of 6 and 11 years of their children. Furthermore, mothers are administered the Diagnostic and Statistical Manual of Mental Disorders, fifth edition-TR Self-Rated Level 1 Cross-Cutting Symptom Measure[30] when their children reach 11 years of age. Other repeated maternal measures include lifestyles (eg, body mass index, diet, consumption of alcohol and tobacco), sleep habits, social environment,

quality of life (the 36-Item Short Form Health Survey[31]), health-related quality of life,[32] home environment (the Confusion, Hubbub, and Order Scale[33]), parenting experiences (the Parenting Stress Index short form[34]; the Swedish Parenthood Stress Questionnaire,[35] and general living situation details, and so on). Regarding biological data, as illustrated in figure 1, the BASIC study collected maternal blood samples, microbiota samples, amniotic fluid, umbilical cord biopsy and fetal blood samples at birth. For the description of biological data collection, handling and storage, see the BASIC cohort profile.[18]

Children's outcome measures include general physical measures, sleep behaviour, social–emotional development, behaviour, language development, neurodevelopment, school participation and performance, and more (eg, diet and free time activities, and so on; table 2). General physical measures include growth data collected at birth, 18 months, 6 years, and 11 years. Sleep behaviours measured at 18 months of age focus on a child's sleep habits or problems. Evaluations of the relationship with parents are scheduled at the age of 18 months, 6 and 11 years. Children's social skills

**Table 2** Content of child outcome assessments and maternal surveys included in the U-BIRTH study

| Age of the child | 18 months | 6 years | 11 years |
|---|---|---|---|
| Domain of measure | | | |
| Child outcome assessments | | | |
| General measure | Child's growth, traits, and behaviour*; general Screening† | Child's growth, traits, and behaviour*; general Screening† | Child's growth, traits, and behaviour*; general Screening† |
| Sleep | Sleeping habits and problems* | | |
| Social–emotional development | Relationship with parents* | Child's social skills* | Child's social skills and relationship* |
| Behaviour | CBCL (1.5); ECBQ | CBCL (1.5–5) | CBCL[6–18] |
| Language development | CBCL-LDS General Screening† | Child's language development* General Screening† | Child's language health* |
| Neuropsychological development | | FTF | FTF |
| School | Participation in preschool (parent-report)* | School performance (parent-report)* | Child's school performance (parent-report* and register data‡) |
| Biological sample | Saliva sample | Saliva sample | |
| Others | Free time activities* | Child's chores*; child's free-time* | Child's chores*; child's free-time* |
| Maternal surveys | | | |
| Depressive symptoms | DSRS | PHQ-9 | PHQ-9; DSM-5 Self-Rated Level 1 CCSM |
| Trauma and stressful events | SLE; SPSQ | New SLE | New SLE |
| Medical information | Current medication* Medical journal | Current medication* Medical journal | Current medication* Medical journal |
| Lifestyles | Alcohol*; smoking*; tobacco* | Alcohol*; smoking*; tobacco* | Alcohol*; smoking*; tobacco* |
| Sleep | Sleep quality* | Sleep quality* | Sleep quality* |
| Others | CHAOS; chores and activity*; life right now*; PSS; SF-36; second parent's status*; social life* | Another pregnancy*; CHAOS; chores and activities*; life right now*; maternal leave*; partner's status*; previous breastfeeding*; PSI; SES*; SF-36; social life* | CHAOS; EQ-5; PSQ; SOC; SES* |

Period of usage in the BASIC is stated in parentheses if other than 2009–2017.
For a full list of assessments from pregnancy to 11 years after birth, see online supplemental table 1.
*Questions designed by the research team.
†Data source: the National Board of Health and Welfare https://www.socialstyrelsen.se/. Information includes the records related to referral and visit to a speech therapist at age of 2, 3, 4, and 5 years.
‡Data source: the Swedish National Agency for Education (skolverket) https://www.skolverket.se/skolutveckling/statistik.
CBCL, Child Behaviour Checklist; CBCL-LDS, Child Behaviour Checklist, Language Development Survey; CHAOS, Confusion, Hubbub and Order Scale; DSM-5, the Diagnostic and Statistical Manual of Mental Disorders, the fifth Edition; DSRS, Depression Self-Rating Scale; ECBQ, Early Childhood Behaviour Questionnaire; EQ-5D, Health-Related Quality of Life Questionnaire; FTF, Five to Fifteen; PHQ9, Patient Health Questionnaire; PSI, Parenting Stress Index; PSQ, Parenting Style Questionnaire; PSS, Perceived Stress Scale; SES, socioeconomic status; SF-36, The 36-Item Short Form Health Survey; SLE, Stressful Life Events Scale; SOC-29, Sense of Coherence; SPSQ, Swedish Parenthood Stress Questionnaire.

are measured at 6 and 11 years old. Parent-reported behavioural measures are obtained at 18 months (the Early Childhood Behaviour Questionnaire,[36] the Child Behaviour Checklist for 1.5–5 years old[37]); and 6 years (the Child Behaviour Checklist for 1.5–5 years old[37]) and 11 years of age (the Child Behaviour Checklist for 6–18 years old[38]). Language development is measured at 18 months, 6 years (the Language Development Survey[39]), and 11 years of age. The Five to Fifteen questionnaire,[40 41] is applied to identify symptoms or problems related to attention deficit hyperactivity disorder at the age of 6 and 11 years. The survey also covers daily life, including

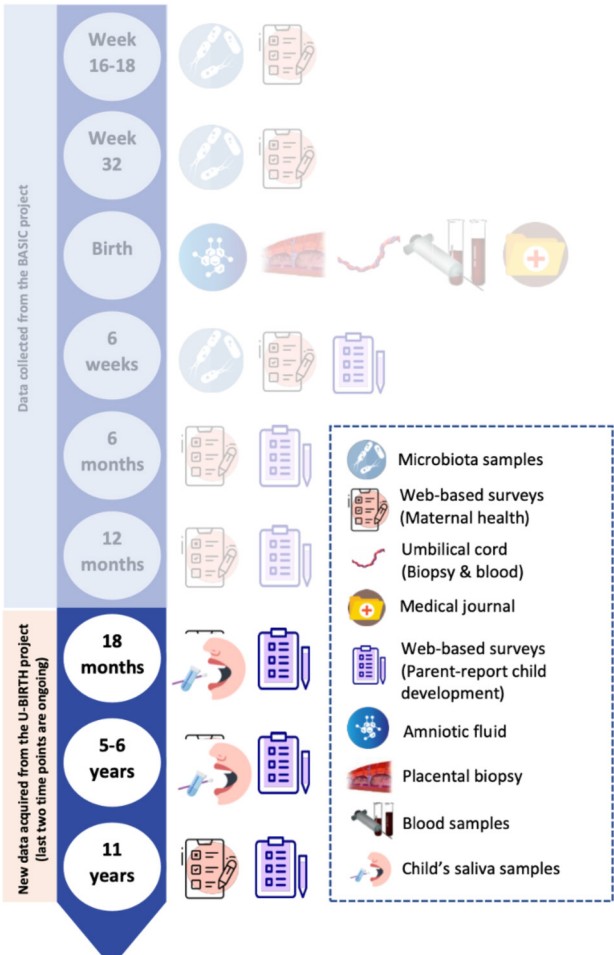

**Figure 1** U-BIRTH study timeline.

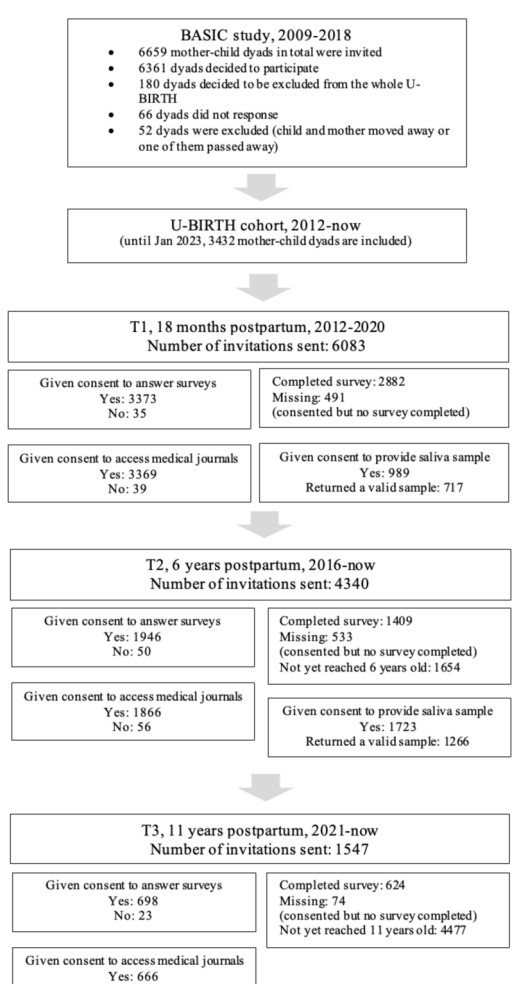

**Figure 2** Flowchart of the U-BIRTH study participants. BASIC, Biology, Affect, Stress, Imaging and Cognition.

school participation, household activities and free time activities, at 18 months, 6 and 11 years of age. Additionally, with parental consent to link data to national register, we collect information on maternal diagnoses, children's diagnoses, child growth, medical referrals, breastfeeding information during the first years post partum for further analysis. Importantly, with parents' consent to access the information from the medical journals and register data, we are able to identify children at the age of 11 years with diagnoses (eg, attention deficit/hyperactivity disorder or autistic spectrum disorder) and potentially establish predictive models. Moreover, a subset of the participants was assessed from pregnancy to 30 months post partum in collaboration with the Uppsala Child and Baby Lab at the Department of Psychology at Uppsala University (the BASIC Child Project, for the project descriptions, https://nyu.databrary.org/volume/828,[42]) between 2014 and 2018. Infants' cognitive and social–emotional development was assessed using behavioural observations, eye-tracking measures and parent reports. Inhibitory control, executive functions, working memories and parental scaffolding behaviours were evaluated by behavioural observations. Data of this subset will also be included in the analysis.

## Patient and public involvement

Patient and public involvement are essential to the current cohort profile. From pilot design to long-term follow-ups, a high degree of engagement from patients and the public continues to shape the development of this study. Our research team has also established open communication through a variety of platforms with the public from the beginning of the study. Study participants have been encouraged to contact the team regarding any comments and questions related to procedures. We have reviewed patients' and the public's feedback and we will continue to take this valuable information into account when examining our research approach and acquired data. The research team has close contact with several stakeholders, including clinical staff and patient organisations.

## FINDINGS TO DATE
### What have we learnt about peripartum depression correlates from the BASIC study?

The BASIC study aimed to identify depression and anxiety symptom patterns and their biophysiological correlates

from pregnancy to 12 months post partum (for the full publication list: https://www.basicstudie.se/publicerade-arbeten). In this cohort, we characterised five distinct trajectories of peripartum depression based on symptom onset: (1) healthy (60.6%), (2) chronically depressed (14.6%), (3) depressed with early postpartum onset (10.9%), (4) depressed only in pregnancy (8.5%), and (5) depressed with late postpartum onset (5.4%).[16] Interestingly, these groups exhibited significant variations in background and psychosocial characteristics. For instance, all trajectories were associated with smoking prior to pregnancy, migraine, premenstrual mood symptoms, intimate partner violence, interpersonal trauma, negative delivery expectations, pregnancy nausea and symphysiolysis. Early postpartum onset was linked to nulliparity, instrumental delivery, or a negative delivery experience, while early and late postpartum onset, along with chronic depression, were associated with other postpartum factors like low partner support, and bonding difficulties. Furthermore, we demonstrated that, in healthy pregnancy, most inflammatory markers increased in the postpartum period.[43] Intriguingly, several markers such as monocyte chemotactic protein 4 tumour necrosis factor ligand superfamily member, hepatocyte growth factor, interleukin, fibroblast growth factor 23, and C-X-C motif chemokine 1, and signal transducing adaptor molecule-binding protein (STAM-BP) were related to different depression trajectories.[44–47] Compared with healthy women, women with antenatal depression or with selective serotonin reuptake inhibitor (SSRI) treatment showed lower levels of peripheral inflammatory markers.[48] When examining metabolic profiles in our cohort, there were no significant depression-specific differences in untargeted gas chromatography-mass spectrometry between women with depressive symptoms and those without depressive symptoms during the antenatal phase.[49] However, five metabolites (glycerol, threonine, 2-hydroxybutanoic acid, erythritol and phenylalanine) were higher among women with postpartum depression than controls.[50] Moreover, pregnancy is associated with a dysregulation of the hypothalamic–pituitary–adrenal axis and an increase in basal cortisol levels. Though the endocrine changes are prominent during pregnancy, there is no significant difference between women with depressive symptoms and those without depressive symptoms,[51] the difference became significant post partum.[52] Intriguingly, among women who did not have depressive symptoms during pregnancy, the levels of corticotropin-releasing hormone in the second trimester were positively correlated with the levels of postpartum depressive symptoms.[53] Additionally, while evidence suggests that placental glucocorticoid receptors are affected by maternal depression or anxiety,[54 55] results from one of our subsets contrast with some previous reports.[56 57] At the gene level, single nucleotide polymorphisms in the hydroxysteroid 11-beta dehydrogenase 2 gene, a key hypothalamic–pituitary–adrenal axis gene, are significantly linked to depressive symptoms in 6 weeks post partum and the stress levels during pregnancy.[58] Our

data showed that women with SSRIs treatment exhibited higher levels of serotonin receptor 1A and neuropeptide Y2 receptor in the placenta compared with healthy women or women with untreated depression.[59] Overall, the BASIC study provides evidence that can support the identification of individual risk profiles. Several psychosocial and emotional risk factors are consistent with previous literature.[28] We have also gained a deeper understanding of several biological and psychological factors that are important for the clinical implications. For instance, biological factors, including corticotropin-releasing hormone,[53] cortisol levels,[52 60] inflammatory/anti-inflammatory markers,[43 47 48] stress-related genetic polymorphisms[58] and plasma metabolic profiling,[50] might be related to different timing of the observed depressive symptoms which can support the diagnosis, treatment and further pathophysiological investigation of peripartum depression. Moreover, our evidence suggests that preventive measures during delivery should be taken to avoid anaemia, negative experience of delivery, and severe obstetric lacerations in order to reduce the risk for peripartum depression.[61 62] For psychological factors, our evidence suggests that when screening peripartum depression, combining evaluation of attachment and neuroticism/trait anxiety[63 64] might facilitate early identification. Currently, several ongoing analyses focus on the integration of biopsychosocial factors using conventional statistical approaches as well as machine learning techniques. Our results will provide evidence to support the development of accurate prediction as well as personalised preventive intervention and treatment.

### What is the impact of peripartum depression on an infant's outcome?

The interplay between maternal well-being, peripartum exposure and childhood environment contributes to the cognitive, social and emotional development of the child.[19 20] The BASIC cohort underscored many important factors and potential underlying mechanisms behind peripartum depression. These findings have further laid the foundation to investigate the associations between peripartum depression and children's outcomes.

From the BASIC Child longitudinal subset, we observed that among mothers with lower levels of maternal postpartum depression, infants showed more gaze-following behaviour at 10 months.[65] We also observed that a higher level of exposure to interpersonal traumatic events moderated the negative impact of anxiety during the second trimester on an infant's attentional control from 6 to 18 months.[66] In the developmental literature, gaze-following is an ability to synchronise visual attention with others[67] and this ability is linked to social learning and emotional regulation in the early years.[68 69] Attentional control in infancy has been postulated as fundamental to later learning and cognitive development.[70] The negative impact of peripartum depression on infants' development was validated in our recent publication using U-BIRTH data at 18 months. We investigated the

associations between different peripartum trajectories and an infant's behavioural problems at 18 months. Our results demonstrated that compared with infants of healthy mothers, infants of depressed mothers showed a higher risk of behavioural problems. Maternal antenatal and persistent depression were associated with a higher degree of behavioural problems and girls were affected to a greater degree. When we analysed the mediating effect of maternal postpartum bonding, maternal persistent and postpartum depression showed a high level of indirect effect on infants' behavioural problems, while antenatal depression had a stronger direct effect.[71] We are currently investigating how maternal depression and anxiety symptoms during pregnancy are associated with infant temperament trajectories (manuscript submitted). This investigation points to a specific importance for maternal anxiety symptoms during pregnancy that are associated with infant temperament being experienced as increasingly difficult throughout the first 18 months. This is in comparison with depressive/anhedonia symptoms that are associated with infant temperament being stably medium–high in difficulty. Also, sex-specific effects were found with girls being more vulnerable towards maternal anxiety during pregnancy and boys towards maternal postpartum depression. Taken together, results from our subset (BASIC Child) and data from U-BIRTH support the hypothesis that there are negative influences of peripartum depression on an infant's outcome at the behavioural level.

At the biological level, several studies reported a significant impact of peripartum depressive symptoms on child biological and emotional outcomes. For instance, we looked at DNA methylation in the cord blood of newborns, which is a form of pregenetic modification and most of the changes happen during embryogenesis.[72] According to the fetal programming hypothesis,[73] it is assumed that DNA methylation might be a possible underlying mechanism that contributes to cross-generational transmission. In our subset with 373 mother–infant dyads, we observed the differential DNA methylation in infants with mothers combined peripartum depressive symptoms and anxiety symptoms or mothers with SSRIs treatment compared with infants with healthy mothers.[74] In our recent, we used DNA extracted in cord blood at birth and examined if DNA methylation differences could affect an infant's behavioural problems at 18 months while taking peripartum depression into account. However, we did nt find evidence to support the fetal programming hypothesis.[75] At the physical level, we investigated the relationship between prenatal depression, the mother's body mass index (BMI) and infant birth weight. Large birth weight is associated with risks for several long-term adverse outcomes, such as obesity, type II diabetes and cardiovascular disease.[76] The main results of our study showed that maternal BMI during the first trimester modulates the association between antenatal depressive symptoms and birth weight. Consequently, in mothers who had a higher BMI and depressive symptoms during pregnancy, their infants were born with a greater body weight compared with infants with healthy mothers with higher BMI or depressed mothers with normal BMI.[77] Previously, we also demonstrated that in mothers without diabetes, placental gene expression in large gestational birth weight infants influences lipid metabolism, including insulin-like growth factor binding protein 1 and leptin.[78] Notably, low leptin levels have been associated with postpartum depressive symptoms.[79] Although the exact mechanisms linking peripartum depression, BMI, and child outcomes need further research, our findings offer valuable insights into the early identification of high-risk groups for prevention programmes and facilitate a deeper understanding of the intricate relationship between depression and obesity. Furthermore, previous evidence has shown that skin-to-skin contact might help the infant regulate stress.[80] However, mothers with peripartum depressive symptoms during pregnancy showed a delayed first breastfeeding session compared with mothers without depressive symptoms. Mothers who experienced depressive symptoms during pregnancy and had a delayed initiation of breast feeding after birth continued to provide less exclusive breastfeeding at 6 weeks post partum.[81] Our results support the notion that both infants and depressed mothers might benefit from targeted breastfeeding support during the first hours after delivery.

### Strengths and limitations

This project has one of Scandinavian's largest longitudinal cohorts, providing comprehensive and rich data across pregnancy, infancy, childhood and preteen phases. To disentangle the complex interaction between maternal well-being, peripartum exposure and childhood environment and how they contribute to the cognitive, social and emotional development of the child, this study includes multidimensional data from biological, psychometrical and behavioural measures. However, several limitations may limit the generalisability of the findings. First, despite including a relatively large sample, the acceptance rate at the baseline of the U-BIRTH cohort from the BASIC cohort was low (ca. 50%). Second, from the participants' sociodemographic information, most parents enrolled hold a degree in higher education and this has been an indication of a low-risk profile.[82] This might be attributed to the study being conducted in a university town, which limits the diversity of the sample. Third, assessments of maternal psychometric data were primarily self-report questionnaires. This approach resulted in a lack of important objective information relating to maternal cognitive functioning and IQ, which could have an impact on, children's outcomes.[83 84] Fourth, mothers experiencing depressive symptoms might have a higher dropout rate compared with mothers without depressive symptoms. Similarly, the cohort's representability might decrease over time due to the increased rate of missing data and decline in participation. Consequently, our data might not fully capture data on the exact severity level

and the possible fluctuation of depressive symptoms over time after the peripartum period.

With regard to outcome measures, though assessments of child development included in the project are well accepted and widely applied in the field, most of them rely heavily on parent-report questionnaires. This could introduce information bias. Moreover, our study only recruited Swedish-speaking mothers. This admittedly excludes many families with an immigrant background who might have other risk factors, such as lack of social support from extended family, language barriers and so on. This could also limit the generalisability of our results. Finally, our project covers a wide range of information, including socioeconomic background, living conditions, maternal health conditions, various lifestyle factors and so on. When analysing data and interpreting our results, we shall very carefully consider the potential heterogeneity present in the dataset. This entails recognising and accounting for the diverse characteristics within our cohort, as they may influence the outcomes and conclusions drawn from our study.

## COLLABORATION

Researchers interested in collaboration may contact the corresponding author, Alkistis Skalkidou, with their request. The U-BIRTH study steering committee considers the scientific quality of the aims and methods of incoming requests in addition to the volume of the requested data or samples. The committee prioritises study questions on peripartum depression and child development.

**Acknowledgements** We are very grateful to all families who participated in this project, as well as to Carolina Wikström, Elisabeth Hansson and Amanda Olsson for administrative support.

**Contributors** AS conceptualised and designed the study. EF, TKK and MR contributed significantly to the design and execution of the project. UE and H-FT contribute to the continuation of the project. H-FT drafted the article, and all authors revised the draft critically for intellectual content and approved the final version. AS is responsible for the overall content as the guarantor.

**Funding** This project is supported by the Uppsala Region (ALF-965886 and ALF-965905), the Marianne and Marcus Wallenberg Foundation (MMW2011.0115), the Göran Gustafsson Foundation (Nr. 1551 A-2015), and the Swedish Research Council (523-2014-07605) to AS as well as the Gillbergska Foundation to EF (year 2022).

**Competing interests** None declared.

**Patient and public involvement** Patients and/or the public were involved in the design, or conduct, or reporting, or dissemination plans of this research. Refer to the Cohort description section for further details.

**Patient consent for publication** Consent obtained from parent(s)/guardian(s).

**Ethics approval** This research project fulfils General Data Protection Regulation requirements for data processing, storage, and security. Ethical approval has been obtained from the Swedish Ethical Review Authority (the Regional Ethical Review Board in Uppsala, reference number 2019/01170, with amendments, and reference number 2022-03146-02). Participants are informed of the study's objectives, and their data confidentiality and security are assured. Consent is obtained before each data collection, with the assurance that participants can withdraw at any time without providing a reason. Research findings will be continuously disseminated through international peer-reviewed journals, the project's website, social media channels, and national and international conferences.

**Provenance and peer review** Not commissioned; externally peer reviewed.

**Data availability statement** Data are available upon reasonable request. Researchers interested in collaboration may contact AS (corresponding author) with their request to be considered by the U-BIRTH study steering committee.

**ORCID iD**
Hsing-Fen Tu http://orcid.org/0000-0003-1787-3548

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
