## [Reviewer comments · BMJ Open]

ARTICLE DETAILS

TITLE (PROVISIONAL)	Cohort profile: the U-BIRTH study on peripartum depression and child development in Sweden
AUTHORS	Tu, Hsing-Fen; Fransson, Emma; Kunovac Kallak, Theodora; Elofsson, Ulf; Ramklint, Mia; Skalkidou, Alkistis

VERSION 1 – REVIEW

REVIEWER	Tommaso Trombetta University of Turin Department of Psychology
REVIEW RETURNED	15-Apr-2023

GENERAL COMMENTS	I would like to thank the editor for the opportunity to review this interesting manuscript and the authors for their efforts. This is an important project that has already achieved interesting results, even given the large sample size, and may provide further important information. Therefore, I would like to congratulate the authors for their extensive work. I have only a few minor revisions that I list below: - In the introduction, I suggest further discussion of the possible clinical implications of both projects.- P. 9 lines 53-54 you say, "In this cohort, five different trajectories of perinatal depression were characterized," this seems to me to be an interesting point, I would suggest to elaborate briefly on the trajectories you refer to.- In the first section of the "findings to date," I would suggest elaborating on the clinical implications of the findings that emerged from the basic project- P. 10 lines 48-49: "In another study, we observed that a mother's adverse childhood experience is highly related to depressive and anxiety symptoms throughout pregnancy and the first yearpostpartum." I suggest moving this part to the previous section, not referring to the effects on infant development- P. 12 lines 39-43: "The main result showed that depressive symptoms interact with the mother's BMI. Consequently, in mothers who had a higher BMI and depressive symptoms during pregnancy, their infants were born with a greater body weight compared to infants with healthy mothers or depressed mothers with normal BMI(89)." I would elaborate on the implications of this finding
--

REVIEWER	Melania Severo University of Foggia
REVIEW RETURNED	20-Jun-2023

GENERAL COMMENTS	Dear Editor, first thank you for asking me to review this interesting contribution. The article titled "Cohort profile: the U-BIRTH study on perinatal
--

	depression and child development in Sweden” represents an interesting article describing a study cohort called U-BIRTH, aimed at assessing the impact of maternal perinatal disease exposure on child development, taking into account biological and environmental factors during intrauterine life and infancy and aimed at identifying early predictors of alterations in neurodevelopment and psychological development. I am reporting some comments as follows in order to recommend some implementations. The authors have described a large body of work that represents an update of the BASIC cohort. It is suggested that the novelty aspects of the present work should be better highlighted than previously published work. To make the reading and understanding of the work steps more fluent, the context and relevant dates should be described in more detail, clarify the periods of follow-up recruitment and data collection. The authors should indicate the criteria for inclusion and exclusion from the study, including how participants were recruited. Likewise, to further clarify the different times of data collection, the number of women for each phase of the study should be reported. For example, it would be useful to indicate phase by phase how many women were contacted, how many chose to join, how many met the inclusion criteria and were included, how many were retained over time, and what the relative dropout rates were, along with the reasons for nonparticipation. The number of participants with missing data for each variable of interest should be indicated. This information, some of which has already been described, might be implemented in the flowchart in Figure 2. These same data, stage by stage, should also be described for the children enrolled in the study. Finally, expand the limitations section by indicating additional potential confounding factors (e.g., lack of maternal IQ assessment, chronicity or resolution of maternal symptoms over time, heterogeneity of data, etc.).
--	--

VERSION 1 – AUTHOR RESPONSE

Comments by Reviewer 1

Reviewer: Dr. Tommaso Trombetta, University of Turin Department of Psychology

1. I would like to thank the editor for the opportunity to review this interesting manuscript and the authors for their efforts. This is an important project that has already achieved interesting results, even given the large sample size, and may provide further important information. Therefore, I would like to congratulate the authors for their extensive work. I have only a few minor revisions that I list below:

RESPONSE: Thank you very much for dedicating your precious time to reviewing our manuscript. We appreciate your suggestions and please kindly see our point-by-point responses below. Here, we would also like to address that we made a minor change (“perinatal” to “peripartum”) in the title and throughout the manuscript. After discussion in the group, we believe that peripartum is a more appropriate term. Minor modifications, which do not alter the manuscripts, are made and listed in the end (Point 7). We do so to improve the content and also correct some small errors found during the revision process. In addition, we apologise that the previous submission did not use double space. In

the current revised version, we apply double space, hence, the page and line numbers are very different from the previous version.

2. In the introduction, I suggest further discussion of the possible clinical implications of both projects.

RESPONSE: Thank you for this suggestion. In the introduction, we highlighted the different characteristics of the BASIC and the U-BIRTH projects. In BASIC, the project investigates the biopsychosocial factors related to peripartum depression, while the U-BIRTH project focuses on the longitudinal influences of peripartum depression on child outcome. The BASIC study enhances the fundamental understanding regarding the mechanisms of peripartum depression and the knowledge could support further identification of individual risk profiles and develop personalized early prevention and intervention programs. In the current U-BIRTH project, we attempt to use the multi-level approach (combined biological, psychosocial, and environmental aspects) to gain deep insights into the longitudinal impact of peripartum depression and identify potential risk factors and protective factors for child development. We hope this will provide evidence to the field to improve the well-being and sustainability of vulnerable families. Please see our newly inserted text on this point, on Page 3, Line 16–25; Page 4, Line 1–9. “Peripartum depression, a significant complication that affects many women during pregnancy and/or the first year postpartum (1), affects one in 4-7 pregnancies (2-5) and poses a serious global public health concern. Peripartum depression heightens the risk of maternal self-harm and suicidal acts (6, 7) and has a substantial and long-term impact during the most vulnerable window of opportunity for the human brain development (8). Exposure to peripartum depression during this critical period might result in lasting physical, social-emotional, neurological, and cognitive problems in children (9-13), even into adulthood (14). However, while the long-term associations between peripartum mental health and child outcomes are well-documented (15), the underlying mechanisms remain less understood. Given the high degree of complexity and the non-linear association between the trajectories of maternal depression and children’s development (16, 17), it is challenging to study factors deeply embedded in the temporal, social, cultural, biological, genetic, and physical environment, etc. The U-BIRTH cohort, builds on the BASIC (Biology, Affect, Stress, Imaging and Cognition) project (18), aiming to provide an in-depth understanding of the relationships between types of peripartum depression and the developmental trajectories of children, as well as the possible mechanisms of cross-generational transmission. The BASIC study explored various maternal factors linked to peripartum depression, enhancing our understanding of the mechanisms. The combination with BASIC study and the U-BIRTH cohort study will aid in identifying individual risk profiles and support personalised early prevention and intervention programs.”

3. P. 9 lines 53-54 you say, "In this cohort, five different trajectories of peripartum depression were characterized," this seems to me to be an interesting point, I would suggest to elaborate briefly on the trajectories you refer to.

RESPONSE: Thank you very much for this comment. We had added information in the following revised text “In this cohort, we characterized five distinct trajectories of peripartum depression based on symptom onset: (a) healthy (60.6%), (b) chronically depressed (14.6%), (c) depressed with early postpartum onset (10.9%), (d) depressed only in pregnancy (8.5%), and (e) depressed with late postpartum onset (5.4%) (16). Interestingly, these groups exhibited significant variations in background and psychosocial characteristics. For instance, all trajectories were associated with smoking prior to pregnancy, migraine, premenstrual mood symptoms, intimate partner violence, interpersonal trauma, negative delivery expectations, pregnancy nausea, and symphysiolysis. Early postpartum onset was linked to nulliparity, instrumental delivery, or a negative delivery experience, while early and late postpartum onset, along with chronic depression, were associated with other postpartum factors like low partner support, and bonding difficulties.” to clarify this point (please see Page 13, Line 24–25; Page 14, Line 1–9). This categorization is one of the key independent variables in the U-BIRTH project and our recent publication by Fransson et al. (2020) using completed dataset

(18 months) demonstrated that maternal persistent- and postpartum depression showed a high level of indirect effect on infants' behavioural problems, while antenatal depression had a stronger direct effect. Currently, several ongoing analyses, e.g. language development, have also examined whether the persistent (chronic) prenatal depression has more adverse impact on a child than other trajectories.

4. In the first section of the "findings to date," I would suggest elaborating on the clinical implications of the findings that emerged from the basic project

RESPONSE: We thank you for this suggestion. In the section "Findings to date", we added information to emphasise the potential implications from prediction to personalised preventive intervention and treatment for peripartum depression. We would also like to point out that although the BASIC project has completed data acquisitions, many ongoing analyses are still going on. Please see our revised sentences (Page 15, Line 10–25; Page 16, Line 1–2) "Overall, the BASIC study provides evidence that can support the identification of individual risk profiles. Several psychosocial and emotional risk factors are consistent with previous literature (28). We have also gained a deeper understanding of several biological and psychological factors that are important for the clinical implications. For instance, biological factors, including corticotropin-releasing hormone (53), cortisol levels (52, 60), inflammatory/anti-inflammatory markers (43, 47, 48), stress-related genetic polymorphisms (58), and plasma metabolic profiling (50), might be related to different timing of the observed depressive symptoms which can support the diagnosis, treatment, and further pathophysiological investigation of peripartum depression. Moreover, our evidence suggests that preventive measures during delivery should be taken to avoid anemia, negative experience of delivery, and severe obstetric lacerations in order to reduce the risk for peripartum depression (61, 62). For psychological factors, our evidence suggests that when screening peripartum depression, combining evaluation of attachment and neuroticism/trait anxiety (63, 64) might facilitate early identification. Currently, several ongoing analyses focus on the integration of biopsychosocial factors using conventional statistical approaches as well as machine learning techniques. Our results will provide evidence to support the development of accurate prediction as well as personalized preventive intervention and treatment."

5. P. 10 lines 48-49: "In another study, we observed that a mother's adverse childhood experience is highly related to depressive and anxiety symptoms throughout pregnancy and the first year postpartum." I suggest moving this part to the previous section, not referring to the effects on infant development

RESPONSE: Thank you very much for pointing this out and for improving the flow of text. We have removed this sentence instead to avoid confusion.

6. P. 12 lines 39-43: "The main result showed that depressive symptoms interact with the mother's BMI. Consequently, in mothers who had a higher BMI and depressive symptoms during pregnancy, their infants were born with a greater body weight compared to infants with healthy mothers or depressed mothers with normal BMI (89)." I would elaborate on the implications of this finding.

RESPONSE: Thank you for providing this important comment. We have further elaborated the findings and also provided evidence from our research group regarding potential mechanisms related to the large gestational birth weight (please see Page 18, Line 2–15). Large gestational birthweight elevates risks for several long-term adverse outcomes, such as obesity, type II diabetes, and cardiovascular diseases. Though more research is needed, our results facilitate a deeper understanding of the potential origins of obesity. Please see our revised text "Large birth weight is associated with risks for several long-term adverse outcomes, such as obesity, type II diabetes, and cardiovascular disease (76). The main results of our study showed that maternal BMI during the first

trimester modulates the association between antenatal depressive symptoms and birth weight. Consequently, in mothers who had a higher BMI and depressive symptoms during pregnancy, their infants were born with a greater body weight compared to infants with healthy mothers with higher BMI or depressed mothers with normal BMI (77). Previously, we also demonstrated that in non-diabetic mothers, placental gene expression in large gestational birthweight infants influences lipid metabolism, including insulin-like growth factor binding protein 1 and leptin (78). Notably, low leptin levels have been associated with postpartum depressive symptoms (79). Although the exact mechanisms linking peripartum depression, BMI, and child outcomes need further research, our findings offer valuable insights into the early identification of high-risk groups for prevention programs and facilitate a deeper understanding of the intricate relationship between depression and obesity.”

7. [Additional and minor modifications]:

In this point, we would like to provide additional information regarding small modifications we made in the current manuscript. The changes did not influence the conceptual design and execution of the projects. Please see the minor changes listed below.

(1) “Perinatal” to “Peripartum” throughout the manuscript: Originally, we used the term perinatal period to describe the time windows across pregnancy and one year after child birth. Here, we updated the term to peripartum according to the latest definition from the American Psychiatric Association. (<https://www.psychiatry.org/patients-families/peripartum-depression/what-is-peripartum-depression>) Here, it states “Peripartum depression refers to depression occurring during pregnancy or after childbirth.”

(2) For the child outcome assessments previous in Supplementary Table 1 (previously Table 2 in main document), assessment tools “ICQ” at 6 weeks and “TBQ” at 12 months were both previously written as IBQ. This was a mistake and now we corrected them. Furthermore, due to the limited pages and numbers for tables, we moved the content of assessment tools from the BASIC study to Supplementary Table 1.

(3) For the maternal survey (originally in Table 3), we merged the content with Table 2 due to the limited pages and numbers for tables. Moreover, we added “DSM-5 Self-Rated Level 1 CCMS” and medical journal, which we did not indicate in the previous manuscript. It was a mistake. We would like to address that the data are included. In the current revised version, Table 2 includes all tools for both maternal survey and child outcome assessment. (Page 11, “Content of child outcome assessments and maternal surveys included in the U-BIRTH”). The full list of assessments for both BASIC and U-BIRTH projects can be found in Supplementary Table 1.

Comments by Reviewer 2

Reviewer: Dr. Melania Severo, University of Foggia

1. The article titled “Cohort profile: the U-BIRTH study on peripartum depression and child development in Sweden” represents an interesting article describing a study cohort called U-BIRTH, aimed at assessing the impact of maternal peripartum disease exposure on child development, taking into account biological and environmental factors during intrauterine life and infancy and aimed at identifying early predictors of alterations in neurodevelopment and psychological development. I am reporting some comments as follows in order to recommend some implementations.

RESPONSE: Thank you very much for your effort reviewing our manuscript and providing us very constructive and critical comments and feedback. We have tried our best to address them. Please kindly find below our point-by-point responses. Here, we would also like to address that we made a minor change (“perinatal” to “peripartum”) in the title and throughout the manuscript. After discussion in the group, we believe that peripartum is a more appropriate term than perinatal. Three minor modifications, which do not alter the manuscripts, are made and listed in the end (Point 7). We do so to improve the content and also correct some small errors found during the revision process.

Additionally, we apologise that the previous submission did not use double space. In the current revised version, we applied double space, hence, the page and line numbers are very different from the version you previously reviewed.

2. The authors have described a large body of work that represents an update of the BASIC cohort. It is suggested that the novelty aspects of the present work should be better highlighted than previously published work.

RESPONSE: We highly agree with this valid comment. In the introduction, we have revised and addressed the distinctive attributes of the BASIC and U-BIRTH projects. Specifically, the BASIC project focuses on the biopsychosocial dimensions linked to peripartum depression, whereas the U-BIRTH project aims at the longitudinal impact of peripartum depression on child outcomes. While the BASIC project recruited pregnant individuals, the U-BIRTH project recruited mother-infant dyads and is a follow-up of BASIC participants and their offspring. Through the BASIC study, we deepen the fundamental knowledge of the underlying mechanisms and risk factors for peripartum depression. This will support the identification of individual risk profiles for early preventive intervention. Given the rich and multifaceted information provided by the BASIC project, the current U-BIRTH project has the possibility to provide deeper understanding of potential underlying mechanisms and protective and risk factors for children's neuropsychological development problems, using this detailed cross-generational dataset (Page 3, Line 16–25; Page 4, Line 1–15). Please see our newly inserted text on this point, on Page 3, Line 16–25; Page 4, Line 1–9. "Peripartum depression, a significant complication that affects many women during pregnancy and/or the first year postpartum (1), affects one in 4-7 pregnancies (2-5) and poses a serious global public health concern. Peripartum depression heightens the risk of maternal self-harm and suicidal acts (6, 7) and has a substantial and long-term impact during the most vulnerable window of opportunity for the human brain development (8). Exposure to peripartum depression during this critical period might result in lasting physical, social-emotional, neurological, and cognitive problems in children (9-13), even into adulthood (14). However, while the long-term associations between peripartum mental health and child outcomes are well-documented (15), the underlying mechanisms remain less understood. Given the high degree of complexity and the non-linear association between the trajectories of maternal depression and children's development (16, 17), it is challenging to study factors deeply embedded in the temporal, social, cultural, biological, genetic, and physical environment, etc. The U-BIRTH cohort, builds on the BASIC (Biology, Affect, Stress, Imaging and Cognition) project (18), aiming to provide an in-depth understanding of the relationships between types of peripartum depression and the developmental trajectories of children, as well as the possible mechanisms of cross-generational transmission. The BASIC study explored various maternal factors linked to peripartum depression, enhancing our understanding of the mechanisms. The combination with BASIC study and the U-BIRTH cohort study will aid in identifying individual risk profiles and support personalised early prevention and intervention programs."

3. To make the reading and understanding of the work step more fluent, the context and relevant dates should be described in more detail, clarify the periods of follow-up recruitment and data collection.

RESPONSE: Thank you very much for pointing this out. We have clarified the relevant time windows regarding different recruitment stages (Page 5, Line 20–22; Page 6, Line 5–11; Figure 2). More specifically, in the U-BIRTH study, T1 (18 months) started in 2012 and ended in 2020. T2 started in 2016 and when the last child reaches 6 years old will be in 2025. Similarly, T3 started in 2021 and when the last child reaches 11 years old will be in 2030. Here is the text from Page 5 "The BASIC study (from September 2009 to November 2018), involved 5492 women (in 6478 pregnancies, ca. 22% of all Uppsala County births)." Text from Page 6 is as following "For the U-BIRTH study

(<https://www.ubirth.se/>), initiated in 2012, all the BASIC study participants not opting out in any way were (will be) invited when their children reached (or reach) 18 months, 6 and 11 years of age. The youngest participants were born in April 2019. Data acquisition for 6-year-olds and 11-year-olds is expected to finish in the summer 2025 and 2030. Prior to each time point for assessments, consent from all participating mother-child dyads is (will be) obtained.”

4. The authors should indicate the criteria for inclusion and exclusion from the study, including how participants were recruited.

RESPONSE: Thank you for this comment. It was an omission from our side. We clarified the exclusion and inclusion criteria of the BASIC study, which also apply to the U-BIRTH study (Page 5, Line 22–25; Page 6, Line 1–5→). The U-BIRTH study invited all the BASIC study participants who had not in any way indicated that they wanted to opt-out of the study. Please see the revised text here “During the BASIC study recruitment, all Swedish-speaking women aged 18 years and older with a routine ultrasound at Uppsala University Hospital were invited to participate. Exclusion criteria included age under 18 years, inadequate Swedish proficiency, protected identities, bloodborne infections and/or non-viable pregnancy detected through routine ultrasound. These criteria remain the same in the U-BIRTH study. Women received written information by post and provided their written consent for modalities they wished to take part in (e.g. questionnaires, blood samples, tissue samples, or microbiota samples). Participating women were followed up from gestational week 16–18 through 12 months postpartum.”

5. Likewise, to further clarify the different times of data collection, the number of women for each phase of the study should be reported. For example, it would be useful to indicate phase by phase how many women were contacted, how many chose to join, how many met the inclusion criteria and were included, how many were retained over time, and what the relative dropout rates were, along with the reasons for nonparticipation. The number of participants with missing data for each variable of interest should be indicated. This information, some of which has already been described, might be implemented in the flowchart in Figure 2. These same data, stage by stage, should also be described for the children enrolled in the study.

RESPONSE: We thank you for this important comment. We have elaborated more details regarding the recruitment time windows (see response to comment 3.). Moreover, we provided more detailed information regarding the number of invitations sent, the number of consents received (both answer survey, to access the medical journals, and to provide saliva samples), the number of surveys completed, missing, and the number of participants who are not yet reached the assessment age (Page 6, Line 18–25; Page 7, Line 1–6; Page 11, Figure 2). For those who did not express a wish to be excluded from the study, we have sent/will send an invitation on all three time points. For those who decided to drop out without any external cause (e.g. death of the child or the mother), some provided reasons such as moving away, but most of them did not specify. For those who did not consent or reply to the invitation, we send a reminder but did not collect the information regarding the reason. Please see the revised text “Currently, 3525 families have completed at least one assessment and 52 mother-child dyads dropped out due to external factors (e.g. death of the mother or the child or the family moved away). Overall, all children have reached 18 months of age and the data acquisition of this phase is completed. There were 6083 invitations to the study for the 18 months postpartum, and 3373 mother-infant dyads (55%) consented to participate. Among those consenting to participate in the survey, 85% completed the survey at 18 months postpartum. Seventy-three percent of the children have today reached 6 years of age. There were 4340 invitations sent, and 1409 (32%) mother-child dyads have fully completed the assessments. Moreover, 1547 (25%) children have reached 11 years of age and 624 have completed the assessments. Families not responding to the invitation receive one email reminder. Families who agree to participate but do not answer the survey receive three email reminders. The flowchart for participants is presented in Figure

2. For those who decided to drop out without any external cause (e.g. death of the child or the mother), some provide reasons such as moving away, while others did not specify. For non-respondents, we send a reminder but cannot ascertain the reasons.”

6. Finally, expand the limitations section by indicating additional potential confounding factors (e.g., lack of maternal IQ assessment, chronicity or resolution of maternal symptoms over time, heterogeneity of data, etc.).

RESPONSE: We appreciate this valuable suggestion. We have indicated that several maternal psychometric assessments in our project are self-report questionnaires which might result in lacking important objective information, such as maternal cognitive functions or IQ which are reported to be important confounders. Please see Page 19, Line 6–19 for the newly revised text “However, there are several limitations that may limit the generalizability of the findings. Firstly, despite including a relatively large sample, the acceptance rate at the baseline of the U-BIRTH cohort from the BASIC cohort was low (ca. 50%). However, several limitations may limit the generalizability of the findings. Firstly, despite including a relatively large sample, the acceptance rate at the baseline of the U-BIRTH cohort from the BASIC cohort was low (ca. 50%). Secondly, from the participants’ sociodemographic information, most parents enrolled hold a degree in higher education and this has been an indication of a low-risk profile (82). This might be attributed to the study being conducted in a university town, which limits the diversity of the sample. Thirdly, assessments of maternal psychometric data were primarily self-report questionnaires. This approach resulted in a lack of important objective information relating to maternal cognitive functioning and IQ, which could have an impact on, children’s outcomes (83, 84). Fourthly, mothers experiencing depressive symptoms might have a higher dropout rate compared to mothers without depressive symptoms. Similarly, the cohort’s representability might decrease over time due to the increased rate of missing data and decline in participation. Consequently, our data might not fully capture data on the exact severity level and the possible fluctuation of depressive symptoms over time after the peripartum period.” In addition, we used educational level as a proxy for cognitive functioning. For the chronicity and remission of the maternal depressive symptoms, we will combine questionnaires at T1 (18 months), T2 (6 years), and T3 (11 years), as well as information acquired from medical journals (Page 14, Line 2–9, Table 2 & Figure 2), to try to map out the development of depressive symptoms. The changes of maternal depressive symptoms will be taken into account in our analyses. We have, nevertheless, noted the absence of longitudinal in-depth assessment of maternal mood in the limitations.

7. [Additional and minor modifications]]:

In this point, we would like to provide additional information regarding small modifications we made in the current manuscript. The changes did not influence the conceptual design and execution of the projects. Please see the minor changes listed below.

(1) “Perinatal” to “Peripartum” throughout the manuscript: Originally, we used the term perinatal period to describe the time windows across pregnancy and one year after child birth. Here, we updated the term to peripartum according to the latest definition from the American Psychiatric Association. (<https://www.psychiatry.org/patients-families/peripartum-depression/what-is-peripartum-depression>) Here, it states “Peripartum depression refers to depression occurring during pregnancy or after childbirth.”

(2) For the child outcome assessments previous in Supplementary Table 1 (previously Table 2 in main document), assessment tools “ICQ” at 6 weeks and “TBQ” at 12 months were both previously written as IBQ. This was a mistake and now we corrected them. Furthermore, due to the limited pages and numbers for tables, we moved the content of assessment tools from the BASIC study to Supplementary Table 1.

(3) For the maternal survey (originally in Table 3), we merged the content with Table 2 due to the limited pages and numbers for tables. Moreover, we added “DSM-5 Self-Rated Level 1 CCMS” and medical journal, which we did not indicate in the previous manuscript. It was a mistake. We would like to address that the data are included. In the current revised version, Table 2 includes all tools for both maternal survey and child outcome assessment. (Page 11, “Content of child outcome assessments and maternal surveys included in the U-BIRTH”). The full list of assessments for both BASIC and U-BIRTH projects can be found in Supplementary Table 1.